# Comparison of the Fracture Resistance and Fracture Mode of Contemporary Restorative Materials to Overcome the Offset of Mandibular Implant-Supported, Cement-Retained Crowns

**DOI:** 10.3390/ma14174838

**Published:** 2021-08-26

**Authors:** Salwa Omar Bajunaid, Ibraheem Alshiddi, Lamya Alhomaidhi, Rania Almutairi, Shoq Alolayan, Syed Rashid Habib

**Affiliations:** 1Department of Prosthetic Dental Science, College of Dentistry, King Saud University, Riyadh 11545, Saudi Arabia; ishiddi@gmail.com (I.A.); syhabib@ksu.edu.sa (S.R.H.); 2King Abdulaziz Medical City-National Guard, Riyadh 14611, Saudi Arabia; lmalhmaidhi@gmail.com; 3Border Guard Medical Center, Riyadh 12221, Saudi Arabia; almutairi.rania@gmail.com; 4Sajer Govermental Hospital, Sajer 17658, Saudi Arabia; shoug.alolayan.sa@gmail.com

**Keywords:** dental crown, dental implant, implant offset, lithium disilicate, porcelain fused to metal, zirconia

## Abstract

Background: The purpose was to compare the fracture resistance and the mode of failure of different contemporary restorative materials to restore implant supported, cement-retained mandibular molars. Methods: Two 5 × 10 mm titanium dental implants were mounted in resin blocks and prefabricated titanium and zirconia abutments were connected to each implant. Each implant received forty crowns resembling mandibular first molars. The specimens were divided into four groups (n = 10/group) for each abutment according to the type of material; Group A: porcelain fused to metal crowns; Group B: monolithic zirconia crowns; Group C: zirconia coping with ceramic veneer; Group D: all ceramic lithium disilicate crowns. Specimens were cemented to the abutments, mounted into a universal testing machine, and vertical static load was applied at a speed of 1 mm/min. The test stopped at signs of visual/audible fracture/chipping. Fracture resistance values were analyzed using ANOVA and Tukey’s tests (α ≤ 0.05). The modes of failure were visually observed. Results: A statistically significant difference (*p* < 0.001) of the fracture resistance values among tested groups was found. The group that showed the highest fracture resistance was Group A for both the titanium and the zirconia abutments (3.029 + 0.248 and 2.59 ± 0.39, respectively) while Group D for both abutments (1.134 + 0.289 and 1.68 ± 0.13) exhibited the least resistance. Conclusions: Fracture resistance and fracture mode varied depending on type of restorative material. For both titanium and zirconia abutments, porcelain fused to metal showed the highest fracture resistance values followed by monolithic zirconia.

## 1. Introduction

During the last few decades, osseointegrated dental implants have become an essential part of dental treatment for edentulous and partially edentulous patients. This treatment modality resulted in a dramatic improvement in the quality of life of these patients [1,2,3]. Under SIT conditions, clinicians may expect survival rates for implant-supported prostheses of >80%. Most implants (60%) did not develop signs of peri-implantitis over a 25-year period.

The success rates of dental implants in partially edentulous patients range between 96.6% and 98.5% [4,5,6]. Single implant-retained restorations are known to have several advantages, both aesthetically and functionally [7]. These advantages include, but are not limited to, the reduction of bone resorption around the implant, the preservation of sound adjacent teeth, and better access to oral hygiene between the implant crown and adjacent natural teeth [7,8].

The survival of dental implants depends on successful osseointegration between the implant and host bone and on the implant’s functional loading capacity [9]. This loading capacity should overcome the stresses and forces of mastication and should ensure the harmonious integration of the crown within the dental arch [10].

The choice of the type of connection (screw-retained vs. cement-retained) and final prosthetic material are vital factors for the long-term success of implant-retained and/or supported restorations [11,12,13,14]. Several restorative materials are commonly used for the prosthetic rehabilitation of osseointegrated dental implants [15]. These materials include metal–ceramics, glass–ceramics, and zirconia-based dental restorations [11]. Despite numerous improvements in ceramic materials, these materials exhibit inherent mechanical failure; where cohesive fracture or veneering porcelain chipping are the most frequent forms of failure [16]. Moreover, the inevitable offset of the final restoration owing to the cylindrical nature of the implant fixture vs. the occlusal table of the final crown (particularly in molar restorations) may have a negative effect on the stress distribution within the bone, the abutment or prosthetic screw, and on the fracture resistance of the final crown [17].

Developments in the computer-aided design/computer-aided manufacturing (CAD/CAM) techniques and the introduction of high strength all-ceramic materials (including monolithic or manually veneered ceramic-based or zirconia-based systems) have proven to be an effective means to strengthen implant retained/supported restorations [18,19]. Nevertheless, in addition to the method of attaching the prosthetic crown to the titanium/zirconia implant abutments via cementation or screw retention, an additional prosthetic concern is the inherent physical properties of the final restorative material [20,21]. 

When considering the final prosthesis of an implant-retained/supported reconstruction, there is insufficient scientific evidence for using a single type of prosthetic material that fulfils all of the requirements to achieve successful restoration; thus, the selection of prosthetic material remains controversial [22].

The exploration of physical properties (e.g., fracture resistance and mode of failure) of commonly used prosthetic materials cemented over titanium or zirconia abutments to restore implant retained/supported crowns will help generate understanding of the behavior of these materials under load. Therefore, the aim of this in vitro study was to compare the fracture resistance and mode of failure of common restorative materials that are used to restore cement-retained implant crowns by replacing mandibular molars with an inevitable offset. The null hypothesis was that there would be no difference in the fracture resistance or fracture mode between the tested restorative materials.

## 2. Materials and Methods

### 2.1. Implant Mounting and Metal Coping Fabrication

A 5 × 10 mm titanium implant (Astra Tech^®^, Dentsply Sirona, Charlotte, NC, USA) was placed in a resin block using a dental surveyor (Figure 1); then, a prefabricated straight titanium abutment was connected to the implant (TiDesign TM, Astra Tech OsseoSpeed, Dentsply Sirona, Charlotte, NC, USA).

For the zirconia abutment, another 5 × 10 mm titanium implant was mounted in a resin block, and a prefabricated zirconia abutment (ZirDesing TM, Astra Tech OsseoSpeed, Dentsply Sirona, Charlotte, NC, USA) was connected to the implant.

The wax pattern of a mandibular first molar was produced using a white milling wax (Ceramill Wax, Amann Girrbach AG, Koblach, Austria) (Figure 2a,b). Then, the wax pattern was cut back and cast into a metal coping. Each implant received four types of restorations that were equally divided into four groups (n = 10/group). The details of the tested crown materials and implant system used are shown in Table 1 (Table 1).

### 2.2. Fabrication of the Groups

#### 2.2.1. Group A: Porcelain Fused to Metal Crowns

The wax patterns for the metal copings were designed using the Exocad CAD/CAM system software (Exocad GmbH, Darmstadt, Germany). Wax milling was performed using a Ceramill Matik milling machine (Amann Girrbach AG, Koblach, Austria). The copings were invested, burnt out, and finally cast using a precious alloy (Wiron^®^ 99, BEGO USA Inc., Lincoln, RI, USA). Porcelain was hand-layered using Vita feldspathic porcelain (Vita, Zahnfabrik, Bad Sackingen, Switzerland).

#### 2.2.2. Group B: Monolithic Full Zirconia Crowns

The crowns were designed using the Exocad CAD/CAM system software followed the by milling of Ceramill Zolid FX multilayer zirconia blocks (Amann Girrbach AG, Koblach, Austria) and firing or sintering of the final crowns.

#### 2.2.3. Group C: Zirconia Copings with Ceramic Veneer

Zirconia copings were designed and milled using the same techniques used for the full zirconia crowns. Then, the copings were veneered by the hand layering technique using IPS e.max Ceram (Ivoclar Vivadent, Liechenstein, Switzerland). 

#### 2.2.4. Group D: Full Contour Lithium Disilicate Crowns

The wax patterns were designed and milled using the same materials and techniques used for the porcelain fused to metal group. Then, the wax patterns were invested and burnt out; then, ingots of the lithium disilicate IPS e.max Press were pressed (Ivoclar Vivadent, Liechenstein, Switzerland) to fabricate the final all-ceramic crowns.

### 2.3. Cementation of Crowns 

All specimens were cemented to the implant abutment using glass ionomer cement (3M™ ESPE™ Ketac™ Cem, Saint Paul, MI, USA). The cement was mixed following the manufacturer’s instructions. Then, the crowns were loaded with a thin layer of the luting cement and were seated with finger pressure. Excess cement was removed, and a minimum of 30 min was given for complete setting before fracture testing.

### 2.4. Fracture Load Testing

The implant resin block was mounted into a universal testing machine (INSTRON 5960 Series Universal Testing Systems, Norwood, MA, USA), and a vertical static load was applied to the center of the occlusal surface of the crowns with a speed of 1 mm/min.

The test stopped when any sign of visual or audible perceptible fracture or chipping occurred. The modes of failure were observed and evaluated by visual analysis. An associated digital software was used to record the breaking load and to display the data as the stress–strain curve. 

### 2.5. Data Analysis

The data were analyzed using the SPSS 20.0 software (SPSS, Chicago, IL, USA) at α ≤ 0.05. One-way analysis of variance (ANOVA) was performed to detect the significant effects of the variables. Tukey’s multiple comparison test was performed to compare the data.

## 3. Results

In the present study, the fracture resistance and mode of failure of common restorative materials used for implant-retained mandibular molars cemented over titanium and zirconia abutments were evaluated and compared. A total of 80 specimens (40 for each abutment groups; and 10 from each restorative material) were tested and evaluated. The means, standard deviations, and ANOVA results of the fracture resistance values of all of the tested groups are shown in Table 2.

Significant variations in the fracture resistance values between the tested restorative materials were observed. One-way ANOVA results confirmed the differences (*p* < 0.000). The highest (3.029 ± 0.248 MPa) and lowest (1.134 ± 0.289 MPa) mean values for the load (MPa) needed to fracture the crowns cemented over the titanium abutment were determined for Group A and Group D. For the crowns cemented over the zirconia abutment, the highest (2.59 ± 0.39 MPa) and lowest (1.68 ± 0.13 MPa) fracture resistance values for the load were also observed for Group A and Group D, respectively.

Multiple comparisons of the fracture resistance values of tested materials are shown in Table 3. The results of the test showed significant differences (*p* < 0.05) in the fracture resistance among the tested restorative materials cemented over the titanium abutments. For the crowns cemented over the zirconia abutments, the difference in the fracture resistance was not significant between Group A and Group B and between Group C and Group D. 

The visual inspection of the tested specimens showed mixed cohesive and/or adhesive fractures. The Group A (metal–ceramic) crowns showed cohesive fracture within the veneering porcelain (Figure 3a and Figure 4a). The Group B (monolithic zirconia) specimens revealed audible failure and cracks for the titanium abutment (Figure 3b) and a bulk fracture (through and through) for the specimens cemented on zirconia abutment (Figure 4b), respectively. Group C (bilayered zirconia) exhibited a through and through fracture from the layered porcelain all the way through the zirconia copings for both types of abutments (Figure 3c and Figure 4c). The Group D (all-ceramic) specimens exhibited catastrophic fractures (bulk) with irregular fracture lines for both abutments (Figure 3d and Figure 4d).

## 4. Discussion

This in vitro research study evaluated the fracture resistance of four restorative materials that are commonly used for the fabrication of implant-supported/retained crowns using a universal testing machine under similar testing conditions. The crowns fabricated from the four restorative materials were cemented over titanium and zirconia abutments. Using computer-aided design (CAD), the tested specimens or crowns were designed to have identical sizes and shapes. This was achieved using cement-retained crowns rather than screw-retained restorations. 

The methodology used in this study for measuring and comparing the fracture resistance (in MPa) of the tested materials (using the Universal Testing Machine software) provides useful values and have been reported in previous research studies [23]. In this study, the single cycle loading technique was employed. This technique is considered acceptable, as has been shown in the literature where no significant difference has been reported between the single cycle loading technique and the mouth motion loading technique in evaluating the fracture resistance of crowns [12,23,24].

The null hypothesis of no difference in the fracture resistance of the tested materials was rejected because significant differences in the mean values of fracture resistance among the tested groups were observed. The results revealed a material-dependent in vitro performance. In this study, any visual or audible perceptible fracture or loss of material in the original crowns was considered to be a sign of failure. However, all of the groups exhibited either a fracture of the entire prosthesis bulk (e.g., in the Group C and Group D specimens) or just a fracture of the veneered ceramic layer (e.g., in the Group A and Group B specimens).

Group A (the control group) showed the highest resistance to fracture over both titanium and zirconia abutments, and this was significantly greater than that of the other tested groups. These specimens showed cohesive failure (i.e., chipping) within the porcelain veneer layer on the cusp tips and a fracture line along the buccal and lingual part of the prostheses. This result agrees with the one in a study performed by Rao et al., where a similar pattern of fracture was observed [24]. The literature showed that ceramic fracture is the third most common reason for failure [18,25]. 

Comparing the fracture resistance of the Group B (monolithic zirconia) and Group C (bilayered, zirconia-based) crowns cemented over titanium and zirconia abutments revealed that, as expected, Group B showed significantly higher fracture resistance values, which only manifested as audible and chipping fractures on the titanium abutment and as core fractures on the zirconia abutment. The fracture patterns of the Group C restorations confirmed mixed cohesive and adhesive fractures. These results could also be attributed to the presence of the junction between the layering or veneering materials and zirconia frameworks, which is a weak bond in bilayered restorations [18]. Moreover, the all-ceramic, monolithic lithium disilicate crowns exhibited the lowest fracture resistance, and bulk fracture was observed in these specimens with irregular asymmetrical fracture lines. This result was in agreement with the findings of de Kok et al., who observed the highest load to fracture for monolithic zirconia crowns followed by lithium disilicate crowns, when the crowns were cemented to a prefabricated titanium abutment [26].

Rao et al. compared the fracture resistance of three different materials of implant-supported crowns and determined that in the bilayered zirconia group, the fracture was only in the veneered layer [24]. This result differs from the present study, which showed that the crowns experienced fracture in both the coping and the veneering layers. A study by Elshiyab et al. compared the fracture resistance between zirconia crowns and monolithic lithium disilicate implant-retained crowns and determined that zirconia crowns had higher fracture resistance [27]. This result was in agreement with the results of this study. Furthermore, Hussien et al. have determined that monolithic zirconia implant-supported crowns exhibited significantly higher fracture resistance than monolithic lithium disilicate and layered zirconia crowns, which was harmonious with the results of this study [28]. This superior ability of zirconia to resist fracture is likely due to the increased crystalline content, fracture toughness, and flexural strength of the zirconia core [23,25]. In contrast, previous studies by Taguchi et al. and Honda et al. did not show significant differences in the fracture resistance between layered zirconia and metal ceramic restorations for cement-retained restorations [29,30].

The primary stability of the implant is crucial for the success of treatment. For the assessment of the primary stability, a novel micromotor probe can be used to measure the bone density at the implant placement site by recording implant insertion torque values and the torque/depth integral. The system provides quantitative data correlating between initial bone-to-implant contact (BIC) and bone density. This information is very critical in selecting the final most resistant restorative material to restore the osseointegrated implant in the above-mentioned clinical cases [31].

Fractographic analysis of interfacial fractures under SEM was done for the porcelain fused to metal and for the zirconia crowns Figure 5a,b.

Some limitations of this study include the known limitations associated with in vitro studies, the absence of a clinical environment, and a small sample size. Future research on other available systems with a larger sample size is recommended. Caution is advised for direct comparison because different test methods, implant systems, restoration designs, loading conditions, cementation materials, and types of metal alloys were used. Within the limitations of this study, there is a need for long-term clinical research to confirm these in vitro results before developing detailed recommendations for the expanded clinical use of such restorations.

## 5. Conclusions

Within the limitations of this study, the authors were able to conclude that there were significant differences in the fracture resistance between the groups tested, where the metal–ceramic crowns showed the highest fracture resistance values; the lithium disilicate all-ceramic group (IPS-empress) was the least resistant to fracture and could not withstand physiological molar masticatory forces. The type of abutments (titanium/zirconia) had no influence on the fracture resistance values of the tested restorative materials. Hence, the authors would recommend metal–ceramic or full zirconia restorative materials to restore implant-supported crowns.

## Figures and Tables

**Figure 1 materials-14-04838-f001:**
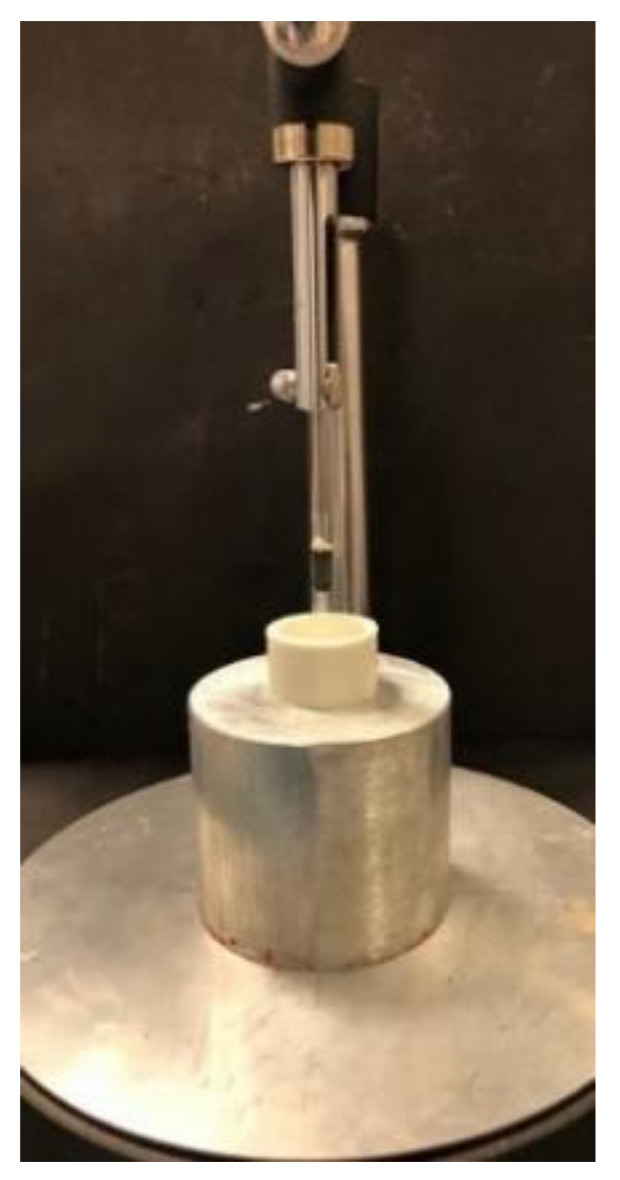
Implant placed in a resin block using a dental surveyor.

**Figure 2 materials-14-04838-f002:**
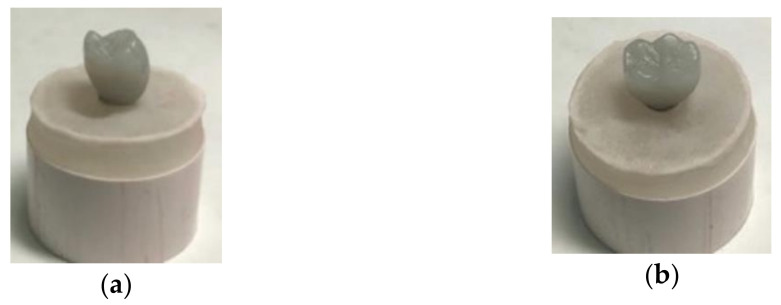
(**a**) Wax pattern of a mandibular first molar; (**b**) occlusal view of the wax pattern of a mandibular first molar.

**Figure 3 materials-14-04838-f003:**
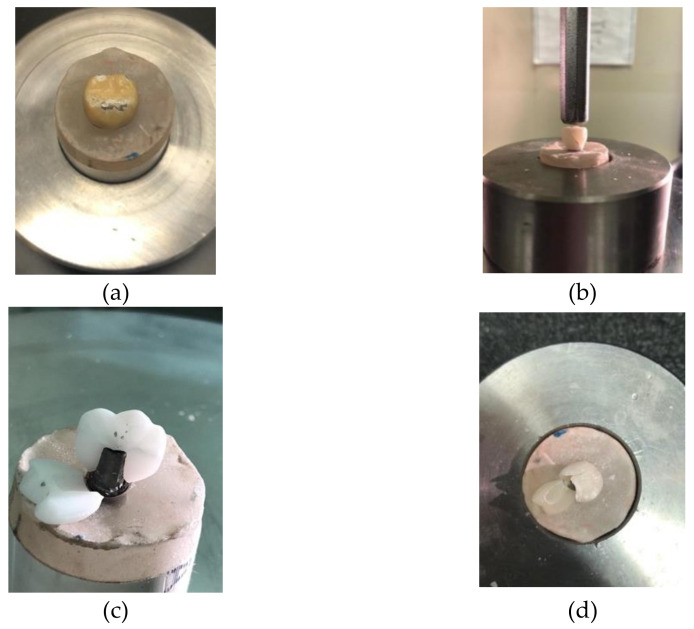
Fractures of the four tested groups on the titanium abutment: (**a**) Group A: porcelain fused to metal; (**b**) Group B: monolithic zirconia; (**c**) Group C: zirconia coping with a ceramic veneer; (**d**) Group D: full contour all-ceramic lithium disilicate.

**Figure 4 materials-14-04838-f004:**
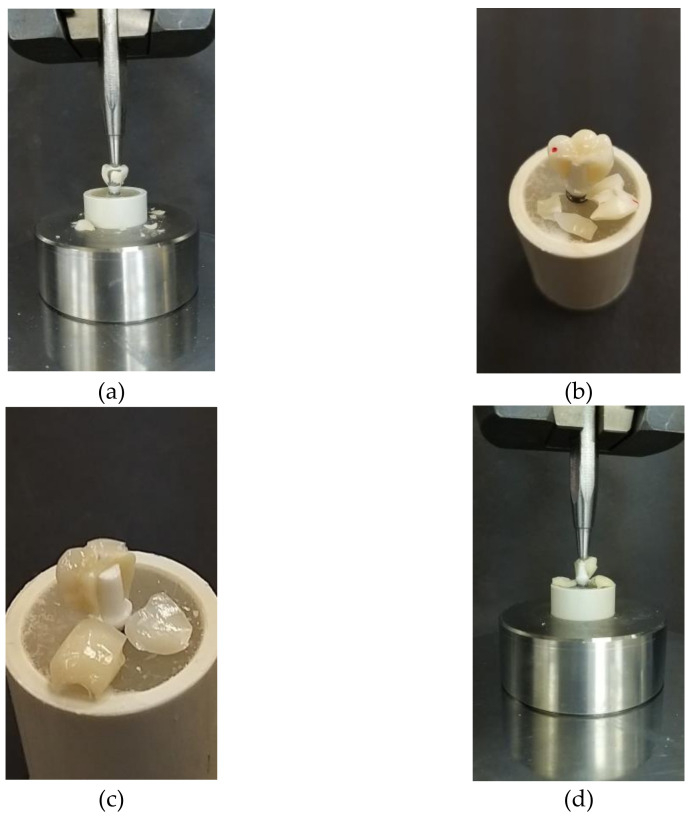
Fractures of the four tested groups on the zirconia abutment: (**a**) Group A: porcelain fused to metal; (**b**) Group B: monolithic zirconia; (**c**) Group C: zirconia coping with a ceramic veneer; (**d**) Group D: full contour all-ceramic lithium disilicate.

**Figure 5 materials-14-04838-f005:**
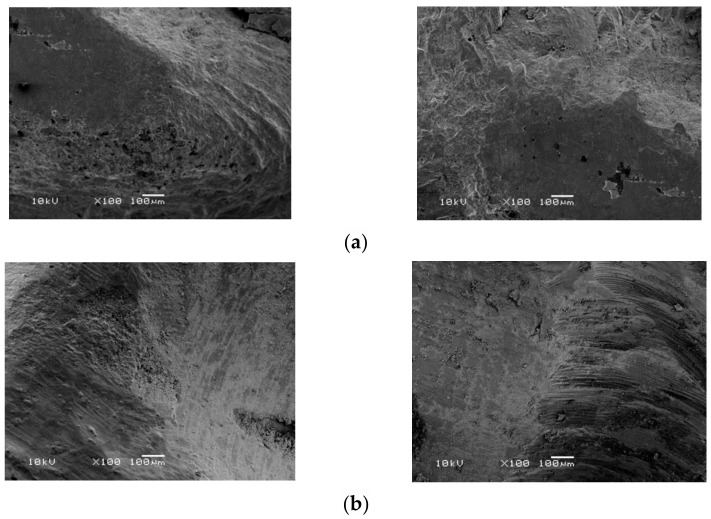
Fractographic analysis under SEM: (**a**) SEM for the porcelain fused to metal sample, (**b**) SEM for the full monolithic zirconia sample.

**Table 1 materials-14-04838-t001:** Details of the implant system and crown materials.

Group	Material	
Group A	Coping	Veneering Ceramic
	Metal coping: Wiron^®^ 99 (BEGO USA Inc. Lincoln, RI, USA)	Vita feldspathic porcelain: (Vita, Zahnfabrik, Bad Sackingen, Switzerland.)
Group B	Monolithic Zirconia: Ceramill Zolid fx multilayer (Amann Girrbach AG, Koblach, Austria)	
Group C	Bilayered Zirconia: Ceramill Zolid fx multilayer (Amann Girrbach AG, Koblach, Austria)	Lithium disilicate: IPS e.max Ceram. (Ivoclar Vivadent, Liechenstein, Switzerland)
Group D	Lithium Disilicate: IPS e.max Press (Ivoclar Vivadent, Liechenstein, Switzerland)	
**Implant System**		**Abutments**
OsseoSpeed^®^ TX (Astra Tech^®^, Dentsply Sirona, Charlotte, NC, USA)		TiDesign^TM^ (Astra Tech^®^, Dentsply Sirona, Charlotte, NC, USA)
		ZirDesign^TM^ (Astra Tech^®^, Dentsply Sirona, Charlotte, NC, USA)

**Table 2 materials-14-04838-t002:** Descriptive statistics with mean, standard deviation, and ANOVA results of the fracture resistance for all tested groups (N = 40).

Abutment	Group	N	* Mean (MPa)	Std. Deviation	ANOVA *p*-Value	95% Confidence Interval for Mean	
						Lower Bound	Upper Bound
Titanium	PFM	10	3.029	0.262	0.000	2.842	3.216
	Monolithic Zirconia	10	2.417	0.341		2.173	2.661
	Layered Zirconia	10	1.955	0.187		1.821	2.088
	Lithium Disilicate	10	1.135	0.305		0.917	1.353
Zirconia	PFM	10	2.59	0.39	0.000	2.310	2.880
	Monolithic Zirconia	10	2.47	0.31		2.253	2.697
	Layered Zirconia	10	1.99	0.40		1.698	2.285
	Lithium Disilicate	10	1.68	0.13		1.594	1.781

* Mean fracture resistance was recorded in mega pascals (MPa).

**Table 3 materials-14-04838-t003:** Multiple comparisons between tested groups by post hoc Tukey’s HSD test.

Abutment	Group	PFM	Monolithic Zirconia	Layered Zirconia	Lithium Disilicate
Titanium	PFM	1	0.000	0.000	0.000
	Monolithic Zirconia	0.000	1	0.004	0.000
	Layered Zirconia	0.000	0.004	1	0.000
	Lithium Disilicate	0.000	0.000	0.000	1
Zirconia	PFM	1	0.850	0.001	0.000
	Monolithic Zirconia	0.850	1	0.012	0.000
	Layered Zirconia	0.001	0.012	1	0.190
	Lithium Disilicate	0.000	0.000	0.190	1

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
