# Peer review of "Comparison of the Fracture Resistance and Fracture Mode of Contemporary Restorative Materials to Overcome the Offset of Mandibular Implant-Supported, Cement-Retained Crowns"

_materials, 2021, doi:10.3390/ma14174838_

Round 1

Reviewer 1 Report

COMMENTS

TABLE 1: change the fonts . It is barely difficult to read

FIGURE 3 : not professionally presented.

LINES 122-124 : could you provide the standards of the tests? How do you think that a static load can reproduce the occlusal forces ?

FIGURE 4 : increase the magnifiction. No fracture is seen or make a picture of the fracture

FIGURE 5 : not professionally presented

CONCLUSIONS : please rewrite the paragraph. It must not be presented as a bullet points.

Reviewer 2 Report

The study is well designed and performed. Topic is of interest since digital, CAD/CAM techniques and monolithic zirconia are more and more in use. But, since there's difference in performance of screw retained crowns and cemented ones this fact should be more emphasised as well in Title and in the text.

Reviewer 3 Report

Thank You for sending article entitled: „ Comparison of the Fracture Resistance and Fracture Mode of Contemporary Restorative Materials to Overcome the Offset of Mandibular Implant-Retained Single Crowns” for the review.

My comments are listed below:

  1. In my opinion results are insignificant, because authors are referring only to other research, claiming their results were the same. That leads to the question – what was the reason for their research? Only to compare with already published? 
  2. Fracture load testing was presented in MPa. Fracture or chipping should be calculated from the formula:

Rt [Pa] = F [N] / S [m2]

where:

Rt - shear stress value

F – force causing failure

S - surface area of the specimen examined

1 MPa = 1,000,000 Pa

What was the surface area of the specimens?

3. There is no explanation where the force was applied.

4. I suggest fractographic analysis of interfacial fractures under SEM, the analysis of chemical composition of the obtained fractures using radiographic spectrum, and adding information about the study in the discussion section.

5. Figures are insignificant. Figure number 3 should be deleted. Figure number should be corrected. Figures should be bigger and focused on samples in order to present analyzed issue instead of the machine.  

Reviewer 4 Report

The authors focus on their study comparing the fracture resistance and fracture mode of contemporary restorative materials to overcome the offset of mandibular implant-retained single crowns. Various crown materials (porcelain fused to metal crowns, monolithic zirconia crowns, zirconia coping with ceramic veneer, full contour, all ceramic lithium disilicate crowns ) were applied to the titanium dental implant, which were then tested.

In the Introduction, the authors describe the need for a dental implant, its aesthetic and functional advantages. The authors report the success rate of dental implants based on studies [4-6] from 1995. It would be appropriate to provide more up-to-date data.

In the chapter Material and methods clearly describes the placement of the dental implant in the test device, the fabrication of crowns according to individual groups, fracture load testing, and data analysis. Currently, 3D printing is increasingly used to produce copings. Do you think the method of manufacturing copings (3D printing, casting) affected the result? On what basis did you determine the speed of the vertical static load?

In Results, the fracture resistance and mode of failure of common restorative materials used for implant-retained mandibular molars were evaluated and compared (10 from each restorative material). In the description Figure 5, you have Group I.-IV., but in the text, you have Group A - D, is that correct?

The Discussion clearly compares the results with the results from other studies.

Finally, the results of the translated study are summarized. The authors could add to the benefit of the study for the dental practice.

Reviewer 5 Report

Dear Editor,

this is well designed and well written manuscript, and I have no comments or suggestions related to it, but the topic is already described in the literature for many times. Maybe to suggest to the authors to resubmit the manuscript to some other lower impacted journal, maybe to Dentistry Journal. if not, they should include zirconia abutments group and compare the group with prefabricated abutments used in the presented study.

Round 2

Reviewer 1 Report

Thanks for the corrections 

Author Response

Thank you!

Reviewer 3 Report

The article is written in an appropriate way.The methods and tools are described with details to allow another researcher to reproduce the results.

The conclusions are interesting for the readership of the Journal. The paper would be attractive for a wide readership.

Author Response

Thank you.

Reviewer 5 Report

Dear authors,

I have suggested to include one more group in your study for this publication, not for new one while the scientific soundness are quite low in presented form.

Author Response

Thank you.